# Is insomnia a risk factor for new-onset asthma? A population-based study in Taiwan

Yu-Chieh Lin,[1] Chih-Cheng Lai,[2] Chih-Chiang Chien,[3] Chin-Ming Chen,[4,5] Shyh-Ren Chiang,[3,5] Chung-Han Ho,[6,7] Shih-Feng Weng,[8] Kuo-Chen Cheng[3,9]

Y-CL and C-CL contributed equally.

For numbered affiliations see end of article.

**Correspondence to**
Dr Kuo-Chen Cheng;
kcg.cheng@gmail.com

## ABSTRACT

**Objectives** To determine whether insomnia at baseline is a risk factor for new-onset asthma.

**Methods** We recruited 48 871 patients with insomnia (insomnia group) newly diagnosed between 2002 and 2007, and 97 742 matched controls without insomnia (control group) from Taiwan's Longitudinal Health Insurance Database 2000. All of the patients were followed up for 4 years to see whether new-onset asthma developed. Patients with previous asthma or insomnia were excluded. The Poisson regression was used to estimate the incidence rate ratios (IRRs) and 95% CIs of asthma. Cox proportional hazard regression was used to calculate the risk of asthma between the two groups.

**Results** After a 4-year follow-up, 424 patients in the insomnia group and 409 in the control group developed asthma. The incidence rate of asthma was significantly higher in the insomnia group (22.01vs10.57 per 10 000 person-years). Patients with insomnia have a higher risk of developing new-onset asthma during the 4-year follow-up (HR: 2.08, 95% CI 1.82 to 2.39). The difference remained significant after adjustment (adjusted HR: 1.89, 95% CI 1.64 to 2.17).

**Conclusions** This large population-based study suggests that insomnia at baseline is a risk factor for developing asthma.

## INTRODUCTION

Insomnia is a sleep disorder that makes falling asleep or staying asleep difficult or prevents people who awaken early in the morning from returning to sleep at least three nights per week for at least 3 months, despite an adequate opportunity for sleep.[1] The prevalence of insomnia is about 25% in Taiwanese adults,[2] and other countries report prevalences ranging from 9% to 50%.[3–5] Insomnia has long been linked with multiple physical conditions, allergies, cancer, hypertension, diabetes, migraine, headache, osteoporosis, fibromyalgia, rheumatoid arthritis, arthrosis, musculoskeletal disorders and obesity.[4] Moreover, impaired sleep is related to developing anxiety and depression.[6–8] It also burdens the public health sector, reduces quality of life and leads to more traffic accidents.[9–15]

**Strengths and limitations of this study**

► The National Health Insurance Research Database (NHIRD) does not contain patient information before 1996.
► NHIRD does not include results of pulmonary function tests or blood tests for inflammatory cytokines, severity levels or the actual duration of insomnia.
► NHIRD does not include some important confounding factors, such as exposure to house dust mites, smoking habits, body mass index, family history or medical compliance.
► Methodology strengths of this study is the large sample size and long-term follow-up which provide considerable statistical power.
► This study can reflect the real-world situation in Taiwan and is more generalisable than are hospital-based or city-based databases.

Asthma is a heterogeneous disease characterised by recurrent attacks of breathlessness and wheezing caused by chronic airway inflammation.[16] Patients with severe asthma may have the problems of insufficient sleep, poor sleep hygiene and clinically significant insomnia.[17 18] Although it is well known that insomnia is more prevalent in patients with asthma,[4 19] Sivertsen *et al*[20] reported that the association might be bidirectional. They did not, however, focus on a single disease, and whether a respondent had asthma was based only on answers from a personal questionnaire rather than on a professional diagnosis, nor did the study adjust for confounding factors. Therefore, we did a prospective cohort study based on clinical diagnoses to determine whether insomnia is a risk factor for new-onset asthma.

## MATERIALS AND METHODS
### Study design and cohorts

Taiwan launched a single-payer National Health Insurance (NHI) programme on

1 March 1995. The NHI databases, one of the largest and most complete population-based datasets in the world, include medical claims information on almost the entire population in Taiwan (>98% in 2009). The data used in this study were taken from the Longitudinal Health Insurance Database 2000 (LHID2000), which contains all claims data from 1996 to 2011 of 1 million (ca. 5% of all enrollees) representative beneficiaries randomly selected from Taiwan's National Health Insurance Research Database (NHIRD) in 2000. The LHID2000 provides encrypted patient identification numbers, sex, date of birth, dates of admission and discharge, the International Classification of Diseases, Ninth Revision—Clinical Modification (ICD-9-CM) codes of diagnoses and procedures, details of prescriptions and expenditure amounts for all outpatient and inpatient medical benefit claims. The institutional review board of Chi Mei Medical Center approved the study and waived the requirement of informed consent because the datasets analysed contained only deidentified patient information.

### Participants

We did a prospective cohort study with two study groups: the insomnia group (patients with newly diagnosed insomnia from 2002 to 2007) and an age-matched, sex-matched and index date-matched control group (patients without insomnia). Patients in the insomnia group (ICD-9-CM codes: 307.41, 307.42, 780.50, 780.52) were diagnosed with insomnia during at least one inpatient hospitalisation or more than three times during outpatient clinic visits within 1 year as previous studies.[21–24] Patients diagnosed with asthma (ICD-9-CM code: 493) before insomnia were excluded.

The flowchart of study subjects selection presented in figure 1. Each patient with insomnia was age-matched, sex-matched and index date-matched to two patients without insomnia. The index dates for the insomnia group patients were the dates of their first registration. Those index dates were used to create index dates for each control group patient. To investigate the risks of developing asthma during the follow-up period, we tracked each participant no more than 4 years from their index date until new-onset asthma, death or the end of the 4-year follow-up. The asthma was also defended as patients with asthma diagnosis at least one inpatient hospitalisation or more than three times during outpatient clinic visits within 1 year.

### Definition of comorbidities

The diagnoses of different physical and mental conditions were based on ICD-9-CM codes during at least one inpatient hospitalisation or more than three times during outpatient clinic visits within 1 year before index date. Comorbidities were as follow: hypertension (401–405, 362.11, 437.2), depression (311, 296.2, 296.3, 300.4), anxiety (293.84, 300.0x, 300.10, 300.2x, 300.5, 309.21), allergic rhinitis (477.x), urticaria (708.x), atopic dermatitis (691.x), bronchiolitis (466.11, 466.19, 079.6), sleep apnoea (327.23, 780.51, 780.53, 780.57), cardiovascular disease (stroke and coronary heart disease) (430–438, 410–414).

### Statistical analysis

Pearson's $\chi^2$ test was used to compare differences in the baseline characteristics and comorbid medical disorders between the insomnia and control cohorts. The incidence rate was calculated as the number of asthma cases during the follow-up, divided by the total person-years for each group. Poisson regression was used to estimate the incidence rate ratios (IRRs) and 95% CIs of asthma between the two groups. A Kaplan-Meier analysis of the cumulative incidence rates of asthma was plotted to describe the proportion of patients with asthma, and the log-rank test was used to analyse the differences between the two cohorts. In addition, Cox proportional hazard regression was used to calculate the risk of asthma between patients with and without insomnia during the follow-up period. SAS V.9.4 was used for all statistical analyses. Significance was set at P<0.05 (two sided). The detectable HR of 1.02 between insomnia group and compared controls was estimated at 90% statistical power and the probability of type I error at 0.05.

### RESULTS

The insomnia group contained 48 871 patients and the control group contained 97 742 (table 1). Insomnia was more common in women (59.96%) and in the middle-aged group (35–49 years: 33.29%). Insomnia group patients were more likely to have comorbidities, such as hypertension (HTN) (21.72% vs 15.12%), anxiety/depression (10.15% vs 1.69%), allergic rhinitis (3.56% vs 1.60%), urticaria (2.37% vs 0.97%), atopic dermatitis (0.51% vs 0.31%), sleep apnoea (0.18% vs 0.02%) and

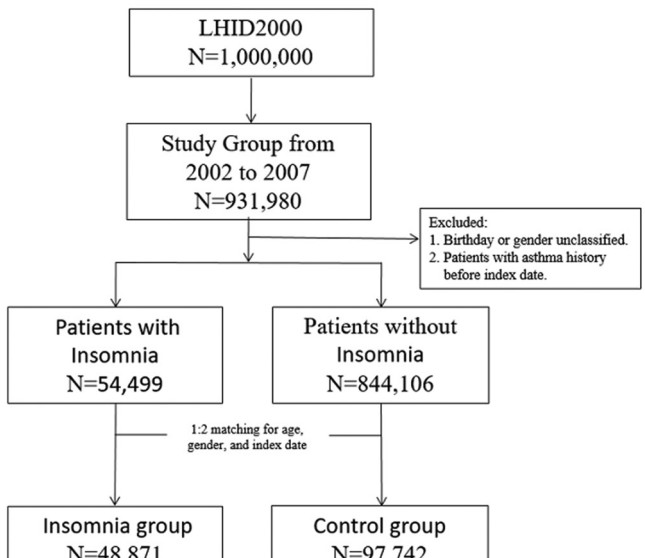

**Figure 1** The flowchart of study subjects selection. LHID, Longitudinal Health Insurance Database.

**Table 1** Baseline demographic characteristics of patients in the insomnia and control groups

| | Insomnia (n=48 871) n (%) | Control (n=97 742) n (%) | P* |
|---|---|---|---|
| **Age group (years)** | | | |
| <34 | 9515 (19.47) | 19 031 (19.47) | 1.00 |
| 35–49 | 16 269 (33.29) | 32 541 (33.29) | |
| 50–64 | 13 611 (27.85) | 27 221 (27.85) | |
| ≥65 | 9476 (19.39) | 18 949 (19.39) | |
| **Sex** | | | |
| Male | 19 569 (40.04) | 39 138 (40.04) | 1.00 |
| Female | 29 302 (59.96) | 58 604 (59.96) | |
| **Comorbidity** | | | |
| Hypertension | 10 616 (21.72) | 14 781 (15.12) | <0.001 |
| Anxiety/depression | 4961 (10.15) | 1649 (1.69) | <0.001 |
| Allergic rhinitis | 1742 (3.56) | 1561 (1.60) | <0.001 |
| Urticaria | 1158 (2.37) | 949 (0.97) | <0.001 |
| Atopic dermatitis | 247 (0.51) | 299 (0.31) | <0.001 |
| Bronchiolitis | 57 (0.12) | 130 (0.13) | 0.44 |
| Sleep apnoea | 87 (0.18) | 21 (0.02) | <0.001 |
| Cardiovascular disease | 4936 (10.10) | 6187 (6.33) | <0.001 |
| **Geographic region** | | | |
| North | 22 674 (46.40) | 46 019 (47.08) | <0.001 |
| Central | 11 144 (22.80) | 17 352 (17.75) | |
| South | 13 803 (28.24) | 32 264 (33.01) | |
| East | 1250 (2.56) | 2107 (2.16) | |
| **SES (monthly insurable wage in NT$)†** | | | |
| <20 000 | 36 415 (74.51) | 69 456 (71.06) | <0.001 |
| 20 000–40 000 | 8370 (17.13) | 18 507 (18.93) | |
| >40 000 | 4086 (8.36) | 9779 (10.00) | |

*P determined using $\chi^2$ tests.
†US$1=NT$30.
NT$, New Taiwan dollar; SES, socioeconomic status.

cardiovascular disease (CVD) (10.10% vs 6.33%) than were control group patients. The insomnia group tended to have lower incomes than did the control group.

After a 4-year follow-up, 424 patients in the insomnia group and 409 patients in the control group had developed asthma. The incidence rate of asthma was significantly higher in the insomnia group (22.01 vs 10.57 per 10 000 person-years; IRR: 2.08 (95% CI 1.82 to 2.39; P<0.001) (table 2). Patients with insomnia had a higher probability of developing asthma during the 4-year follow-up (HR: 2.08; 95% CI 1.82 to 2.39). This difference was still significant after adjustment (adjusted HR (AHR): 1.89; 95% CI 1.64 to 2.17) (table 3). Advanced age was also related to a higher risk for developing asthma (≥65 years: AHR: 8.13; 95% CI 5.89 to 11.23). Patients with higher incomes were less likely to develop asthma (NT$>40 000: AHR=0.62; 95% CI 0.44 to 0.89). People living in eastern Taiwan were more likely to have new-onset asthma then those living

in other regions of Taiwan (AHR=2.01; 95% CI 1.44 to 2.81). The subgroup analysis showed that insomnia was associated with a greater risk of asthma in patients with HTN and CVD (table 4). A Kaplan-Meier survival curve shows that the insomnia group had a higher cumulative incidence rate of asthma than did the control group (P<0.001) (figure 2).

## DISCUSSION

This is the largest cohort study focused on the association between insomnia and new-onset asthma, and the first one on an Asian population. We found that patients with insomnia who sought medical assistance had a significant risk for developing asthma within 4 years of asking for treatment. In this study, the incidence of asthma in control group was 10.57 per 10 000 person-years. This finding is similar to the most recent epidemiology study[25] in Taiwan

**Table 2** Risks of asthma in the insomnia and control groups

| Characteristics | Insomnia | | | | Control | | | | IRR (95% CI) | P* |
|---|---|---|---|---|---|---|---|---|---|---|
| | No | Asthma | PY | IR† | No | Asthma | PY | IR† | | |
| All | 48871 | 424 | 192623.41 | 22.01 | 97742 | 409 | 387095.85 | 10.57 | 2.08 (1.82 to 2.39) | <0.001 |
| Age group (years) | | | | | | | | | | |
| <34 | 9515 | 27 | 37931.78 | 7.12 | 19031 | 17 | 76049.26 | 2.24 | 3.18 (1.74 to 5.84) | <0.001 |
| 35–49 | 16269 | 82 | 64569.83 | 12.70 | 32541 | 32 | 129858.42 | 2.46 | 5.15 (3.43 to 7.75) | <0.001 |
| 50–64 | 13611 | 85 | 53788.73 | 15.80 | 27221 | 83 | 108051.32 | 7.68 | 2.06 (1.52 to 2.78) | <0.001 |
| ≥65 | 9476 | 230 | 36333.07 | 63.30 | 18949 | 277 | 73136.86 | 37.87 | 1.67 (1.40 to 1.99) | <0.001 |
| Sex | | | | | | | | | | |
| Male | 19569 | 180 | 76611.01 | 23.50 | 39138 | 179 | 154370.93 | 11.60 | 2.03 (1.65 to 2.49) | <0.001 |
| Female | 29302 | 244 | 116012.41 | 21.03 | 58604 | 230 | 232724.93 | 9.88 | 2.13 (1.78 to 2.55) | <0.001 |
| Comorbidity | | | | | | | | | | |
| Hypertension | 10616 | 180 | 41264.43 | 43.62 | 14781 | 167 | 57535.99 | 29.03 | 1.50 (1.22 to 1.86) | <0.001 |
| Anxiety/depression | 4961 | 59 | 19475.68 | 30.29 | 1649 | 15 | 6417.10 | 23.38 | 1.30 (0.74 to 2.28) | 0.37 |
| Allergic rhinitis | 1742 | 19 | 6881.46 | 27.61 | 1561 | 8 | 6157.11 | 12.99 | 2.13 (0.93 to 4.85) | 0.07 |
| Urticaria | 949 | 6 | 3719.76 | 16.13 | 1158 | 6 | 4556.88 | 13.17 | 1.23 (0.40 to 3.80) | 0.73 |
| Atopic dermatitis | 247 | 1 | 973.35 | 10.27 | 299 | 4 | 1152.70 | 34.70 | 0.30 (0.03 to 2.65) | 0.28 |
| Bronchiolitis | 57 | 1 | 227.06 | 44.04 | 130 | 1 | 515.46 | 19.40 | 2.27 (0.14 to 6.29) | 0.56 |
| Sleep apnoea | 87 | 1 | 344.54 | 29.02 | 21 | 1 | 80.21 | 124.70 | 0.23 (0.01 to 3.72) | 0.30 |
| CVD | 4936 | 117 | 18838.57 | 62.11 | 6187 | 114 | 23618.24 | 48.27 | 1.29 (0.99 to 1.67) | 0.06 |

*P values were determined using Poisson regression models.
†IR: per 10000 person-years.
CVD, cardiovascular disease; IR, incidence rate; IRR, incidence rate ratio; PY, person-years.

**Table 3** Cox proportional hazard regressions for the development of asthma during the follow-up period

| Variable | Crude HR (95% CI) | Adjusted HR (95% CI)* |
|---|---|---|
| Insomnia | | |
| No | Ref | Ref |
| Yes | 2.08 (1.82 to 2.39) | 1.89 (1.64 to 2.17) |
| Age group | | |
| <34 | Ref | Ref |
| 35–49 | 1.52 (1.07 to 2.15) | 1.57 (1.11 to 2.22) |
| 50–64 | 2.69 (1.93 to 3.75) | 2.45 (1.75 to 3.43) |
| ≥65 | 11.95 (8.78 to 16.26) | 8.13 (5.89 to 11.23) |
| Sex | | |
| Male | Ref | Ref |
| Female | 0.88 (0.76 to 1.00) | 1.00 (0.87 to 1.15) |
| Comorbidity | | |
| Hypertension | 3.47 (3.02 to 3.98) | 1.24 (1.06 to 1.45) |
| Anxiety/depression | 2.08 (1.64 to 2.65) | 1.19 (0.93 to 1.53) |
| Allergic rhinitis | 1.46 (0.99 to 2.14) | 1.33 (0.90 to 1.96) |
| Urticaria | 1.01 (0.57 to 1.78) | 0.81 (0.46 to 1.44) |
| Atopic dermatitis | 1.64 (0.68 to 3.95) | 1.15 (0.48 to 2.78) |
| Bronchiolitis | 1.88 (0.47 to 7.52) | 2.09 (0.52 to 8.38) |
| Sleep apnoea | 3.29 (0.82 to 13.17) | 2.82 (0.70 to 11.37) |
| Cardiovascular disease | 4.84 (4.16 to 5.63) | 1.80 (1.52 to 2.14) |
| Geographic region | | |
| North | Ref | Ref |
| Central | 1.29 (1.08 to 1.55) | 1.26 (1.05 to 1.51) |
| South | 1.28 (1.09 to 1.50) | 1.18 (1.00 to 1.38) |
| East | 2.45 (1.68 to 3.29) | 2.01 (1.44 to 2.81) |
| SES (monthly insurable wage in NT$)† | | |
| <20 000 | Ref | Ref |
| 20 000–40 000 | 0.27 (0.21 to 0.36) | 0.53 (0.40 to 0.71) |
| >40 000 | 0.35 (0.25 to 0.49) | 0.62 (0.44 to 0.89) |

*Parameters were adjusted for all covariates included in the model.
†US$1=NT$30.
NT$, New Taiwan dollar; Ref, reference value; SES, socioeconomic status.

that the incidence of asthma was 9.8 per 10 000 person-year. In contrast, the insomnia group had significant higher incidence of asthma—22.01 per 10 000 person-year than the control group and the general population in previous study.[25] Several studies have reported linkages between insomnia and chronic illnesses, including asthma. A cross-sectional study[19] of 3283 adults showed a higher prevalence of insomnia in those with asthma (adjusted OR (AOR) 1.6; 95% CI 1.3 to 2.0). A 5-year prospective study[26] of 2316 middle-aged adults reported that patients with insomnia at baseline had a higher incidence of asthma (AOR=17.9; 95% CI 2.28 to 140) than did patients without insomnia. But no causal relationship between insomnia and asthma was supported in these

**Table 4** Stratified analysis for patients with hypertension or cardiovascular disease

| | Patients with HTN (n=25 397) AHR (95% CI) | P | Patients with CVD (n=11 123) AHR (95% CI) | P |
|---|---|---|---|---|
| Control | 1.00 (ref.) | | 1.00 (ref.) | |
| Insomnia | 1.59 (1.28 to 1.97) | <0.001 | 1.38 (1.06 to 1.79) | 0.02 |

AHR, adjusted HR; CVD, cardiovascular disease; HTN, hypertension; ref., reference value.

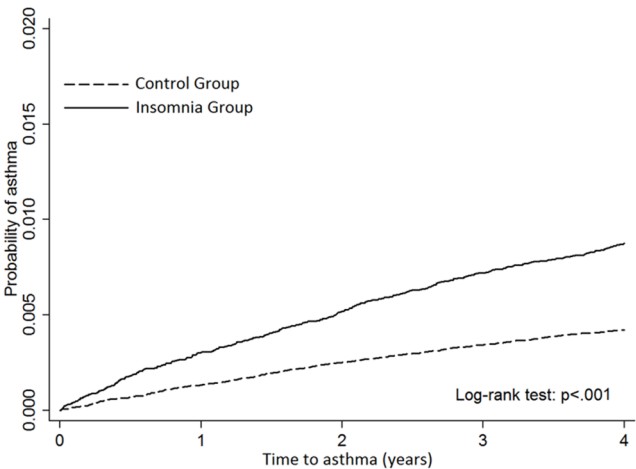

**Figure 2** The probability of developing asthma in insomnia and control group.

two studies. Unlike those two studies, our study excluded patients who had previously been diagnosed with asthma, and we followed our patients for 4 years, which provided evidence that insomnia is a risk factor for developing asthma.

Sivertsen et al[20] reported that insomnia was a significant risk factor for incidence of asthma in an 11-year large population-based prospective cohort study consisted of 24 715 participants (OR=1.47; 95% CI 1.16 to 1.86). Brumpton et al[27] also found that insomnia symptoms were associated with increased risk of incident asthma in the same population-based which consisted of 17 927 participants (three insomnia symptoms, OR=1.70; 95% CI 1.37 to 2.11). However, they used questionnaires to define insomnia and provided no information about the severity or duration of their participants' poor sleep. The diagnoses of asthma and comorbidities were based on self-reports instead of physician-reported diagnoses. Their sample size was smaller than was ours, and the confounding factors adjusted for in their analysis did not include specific asthma or insomnia-related comorbidities such as atopy. In contrast, we used ICD-9 CM codes from Taiwan's LHID2000, which indicated the patients' physician-diagnosed illnesses, including insomnia, asthma and all comorbidities. Besides, we believed that the insomniacs in our cohort suffered from more severe sleep-related symptoms that needed medical assistance. Our findings were statistically significant after our analyses had been fully adjusted for the most common confounding factors related to asthma and insomnia.

Although our findings and those of other studies[17 18 20] indicate a significant association between insomnia and asthma, the pathophysiology of insomnia-related asthma is still unclear. Nonetheless, the common inflammatory pathway between insomnia and asthma should be considered, and several mechanisms have been proposed to explain the potential relationship between insomnia and asthma. First, poor sleep increases interleukin 6 (IL-6) production, and this response lasts until daytime.[28 29] Patients with stable and acute asthma had significantly higher IL-6 production levels in serum, sputum and bronchoalveolar lavage fluid than did healthy controls.[30 31] IL-6 production in the airway promotes allergic airway inflammation in mice. In contrast, using IL-6 knockout mice in the same model showed significantly less mucus secretion.[32 33] Therefore, insomnia might contribute to inducing IL-6 production and that might exacerbate airway hypersensitivity. Second, nuclear factor kappa-light-chain-enhancer of activated B cells (NF-κB) can be induced by sleep loss.[34] Prolonged airway epithelial NF-κB activation has been reported in patients with asthma. Even temporal NF-κB activation in the airway epithelium is sufficient to induce airway hyper-responsiveness in mice.[35] Thus, the higher incidence of asthma in patients with insomnia might be the result of temporal or persistent NF-κB activation induced by sleep loss. Third, insomnia is related to a reduction in interferon gamma (IFN-γ).[36] IFN-γ production, which inhibits airway epithelial inflammation, is lower in patients with asthma than in healthy controls.[37 38] These studies suggest that IFN-γ plays a significant role between insomnia and asthma.

The strengths of our study are its prospective cohort design and that we used the LHID2000, a subset of Taiwan's NHIRD, which contains physician-provided clinical diagnoses instead of illnesses self-reported by patients. The LHID2000 reflects the real-world situation in Taiwan and is more generalisable than are hospital-based or city-based databases. The large sample size and long-term follow-up also provide considerable statistical power.

Our study also has some limitations. First, the NHIRD does not include results of pulmonary function tests or blood tests for inflammatory cytokines, severity levels or the actual duration of insomnia, some important confounding factors, such as exposure to house dust mites, smoking habits, body mass index, family history or medical compliance. Second, we did not assess the association between the medication for insomnia and asthma. However, it would be interesting to note if there were any differences in the development of asthma in those patients with insomnia who were treated versus those who were not. Further study is warranted to investigate the drug effect. Third, some patients might have minor asthmatic symptoms and have not been diagnosed yet before insomnia. Forth, sampling bias, such as a higher number of individuals likely to seek medical care among insomnia group than control group is possible and it may confound the analysis. Finally, because the NHIRD does not contain patient information before 1996, some of our patients might have been misclassified if they were diagnosed with asthma or insomnia before that year.

## Conclusion

We found that patients with insomnia who required medical assistance had a higher risk for developing new-onset asthma. Proper treatments for patients with insomnia might help prevent the progress of airway

inflammation. Additional study is needed to identify the actual mechanism that connects insomnia and asthma.

**Author affiliations**
[1]Department of Family Medicine, Jiannren Hospital, Kaohsiung, Taiwan
[2]Department of Intensive Care Medicine, Chi Mei Medical Center, Liouying, Taiwan
[3]Department of Internal Medicine, Chi Mei Medical Center, Yung Kang, Tainan, Taiwan
[4]Department of Intensive Care Medicine, Chi Mei Medical Center, Tainan, Taiwan
[5]Department of Recreation and Healthcare Management, Chia Nan University of Pharmacy and Science, Tainan, Taiwan
[6]Department of Medical Research, Chi Mei Medical Center, Tainan, Taiwan
[7]Department of Hospital and Health Care Administration, Chia Nan University of Pharmacy and Science, Tainan, Taiwan
[8]Department of Healthcare Administration and Medical Informatics, Kaohsiung Medical University, Kaohsiung, Taiwan
[9]Department of Safety, Health, and Environmental Engineering, Chung Hwa University of Medical Technology, Tainan, Taiwan

**Contributors** Y-CL, C-CL and K-CC designed the study, interpreted the data and drafted and revised the article. C-CC, C-MC and S-RC contributed to interpreting the data and revising the article. C-HH and S-FW contributed to the statistical analysis. K-CC critically reviewed and revised the article. All of the authors read and agreed with the final version of the manuscript.

**Competing interests** None declared.

**Patient consent** Detail has been removed from this case description/these case descriptions to ensure anonymity. The editors and reviewers have seen the detailed information available and are satisfied that the information backs up the case the authors are making.

**Ethics approval** Institutional review board of Chi Mei Medical Center.

**Provenance and peer review** Not commissioned; externally peer reviewed.

**Data sharing statement** The data on the study population that were obtained from the NHIRD (https://nhird.nhri.org.tw/en/) are maintained in the NHRI (http://nhird.nhri.org.tw/). The NHIRD is limited for research purposes only. Applicants must follow the Computer-Processed Personal Data Protection Law (http://www.winklerpartners.com/?p=987) and related regulations of National Health Insurance Administration. All applications are reviewed for approval of data release. Interested researchers may submit queries related to data access to nhird@nhri.org.tw.

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
