## [Reviewer comments · BMJ Open]

ARTICLE DETAILS

TITLE (PROVISIONAL)	Is Insomnia a Risk Factor for New-Onset Asthma? A Population-Based Study in Taiwan
AUTHORS	Lin, Yu-Chieh; Lai, Chih-Cheng; Chien, Chih-Chiang; Chen, Chin-ming; Chiang, Shyh-Ren; Ho, Chung-Han; Weng, Shih-Feng; Cheng, Kuo-Chen

VERSION 1 – REVIEW

REVIEWER	Fredrik Sundbom Uppsala University, Sweden
REVIEW RETURNED	13-Aug-2017

GENERAL COMMENTS	It was showed that having an insomnia diagnosis at baseline was a risk factor for development of new-onset asthma, and that the difference remained significant after adjustment for comorbidity. The paper is structured and follows the STROBE guidelines. The study benefits from a large population and a long observation time. However, I have some concerns, mainly about the methods and the definitions used. Both asthma and insomnia were defined by DSM-9 CM codes. This may work well for asthma, but it may result in a false low prevalence of insomnia, since only individuals with severe insomnia actually have a insomnia diagnosis. Hence, the present study is difficult to compare to previous studies in the same field, of which most use questionnaires on insomnia symptoms. It may also lead to sampling bias, since a higher number of individuals likely to seek medical care may be found in the insomnia group. The lack of data on BMI is a major problem, since obesity is over-represented with insomnia and considered as a key factor in asthma development among adults. If adjustments could have been made for BMI (and, simirlarly, for smoking history) it would have added strength to the results. This very important limitation should at least be more clearly addressed. The results were adjusted for sleep apnea, but the prevalence was surprisingly low in both groups. Adjustments were also made for anxiety and depression, but also these conditions tend to be under-reported using only DSM-9 CM codes. The authors refer to Sivertsen's paper from the Norwegian HUNT study published in 2014. However, I think that another, more recent and highly relevant publication from the HUNT study by Brumpton et al: Prospective study of insomnia and incident asthma in adults: the HUNT study (ERJ, 2017) also should be referred to.
--

REVIEWER	Dr Himender Makker University College London Hospital London UK
REVIEW RETURNED	21-Sep-2017

GENERAL COMMENTS	In this prospective cohort study the authors have investigated insomnia as a risk factors for new onset asthma in Taiwanese population and found that patients with insomnia had almost twice the risk of developing asthma. Approximately 50,000 patients with newly diagnosed insomnia and twice the number of matched control were followed up by four years for new onset asthma. The diagnostic tool used for diagnosis of insomnia is not stated except that it was diagnosed during one hospital admission and three outpatient visits within a year. Insomnia is a syndrome ranging from acute transient sleep disturbance to chronic sleep disruption. Similarly the basis for new onset asthma during follow up is not explained except that it was diagnosed during one hospital admission or three clinic visits. Prevalence of atopy related disorders (allergic rhinitis, atopic dermatitis and urticaria) was significantly higher in insomnia population than control group. This could influence diagnosis of new onset asthma in insomnia population. Authors need to discuss possible effect of above differences in diagnosing new onset asthma-despite Authors need to provide more details for inclusion and exclusion criteria. Authors have not clarified if patients with asthma were actively excluded from case and control group and basis of exclusion. Insomnia is dynamic disease and it might have changed /resolved in insomnia group and might have developed in control group during follow up-were above changes monitored during follow up.
---

REVIEWER	Malvika Sagar Mc Lane Children's Hospital/Baylor Scott and White
REVIEW RETURNED	23-Sep-2017

GENERAL COMMENTS	It is a very large , well written study where authors have addressed a very important question-is insomnia a risk factor for asthma. I have few minor questions related to the paper. 1. Under the material and method section lines 41-44, Patients diagnosed with asthma before insomnia were excluded. It is possible that the patients with symptoms of asthma and not diagnosed yet may have been included. You may add that in limitation of the study. 2. Figure 1: Please consider rephrasing the legend to 'The probability of developing asthma in insomnia and control group'. 3. Sometimes in adult population COPD can be misdiagnosed as asthma and vice versa. The same patient may have been diagnosed with COPD and asthma. Could you look into that?
--

	4. Do you have data on how many patients with insomnia were treated? It would be interesting to note if there were any differences in outcome in those insomnia patients who were treated versus those who were not. This may add weight to your conclusion that proper treatment of insomnia patients might help prevent the progress of airway inflammation. 5. Your sample size is very large. However, for the sake of completeness, did you do the power calculation?
--	--

VERSION 1 – AUTHOR RESPONSE

Reviewer: 1

Reviewer Name: Fredrik Sundbom

Institution and Country: Uppsala University, Sweden

Comment: It was showed that having an insomnia diagnosis at baseline was a risk factor for development of new-onset asthma, and that the difference remained significant after adjustment for comorbidity. The paper is structured and follows the STROBE guidelines. The study benefits from a large population and a long observation time. However, I have some concerns, mainly about the methods and the definitions used.

Both asthma and insomnia were defined by DSM-9 CM codes. This may work well for asthma, but it may result in a false low prevalence of insomnia, since only individuals with severe insomnia actually have a insomnia diagnosis. Hence, the present study is difficult to compare to previous studies in the same field, of which most use questionnaires on insomnia symptoms. It may also lead to sampling bias, since a higher number of individuals likely to seek medical care may be found in the insomnia group.

Reply: We agree with you that ICD-9 CM code methods can only help us identify severe insomnia patient who needs medical help. However, this method has been applied for several previous studies (Ref.). Therefore, this kind of analysis should be acceptable. In addition, due to the method differences, the findings in this study may not be used to compare to previous studies using questionnaires on insomnia symptom. Finally, the issues regarding sampling bias, such as a higher number of individuals likely to seek medical care may be found in the insomnia group were added as one of limitations.

Ref.

Chen, P. J., Huang, C. L. C., Weng, S. F., Wu, M. P., Ho, C. H., Wang, J. J., ... & Hsu, Y. W. (2017). Relapse insomnia increases greater risk of anxiety and depression: evidence from a population-based 4-year cohort study. *Sleep Medicine*.

Huang, C. L. C., Weng, S. F., Wang, J. J., Hsu, Y. W., & Wu, M. P. (2015). Risks of treated insomnia, anxiety, and depression in health care-seeking physicians: a nationwide population-based study. *Medicine*, 94(35).

Chuang, Y. C., Weng, S. F., Hsu, Y. W., Huang, C. L. C., & Wu, M. P. (2015). Increased risks of healthcare-seeking behaviors of anxiety, depression and insomnia among patients with bladder pain syndrome/interstitial cystitis: a nationwide population-based study. *International urology and nephrology*, 47(2), 275-281.

Wu, M. P., Lin, H. J., Weng, S. F., Ho, C. H., Wang, J. J., & Hsu, Y. W. (2014). Insomnia Subtypes and the Subsequent Risks of Stroke. *Stroke*, 45(5), 1349-1354.

Comment: The lack of data on BMI is a major problem, since obesity is over-represented with insomnia and considered as a key factor in asthma development among adults. If adjustments could have been made for BMI (and, similarly, for smoking history) it would have added strength to the results. This very important limitation should at least be more clearly addressed. The results were adjusted for sleep apnea, but the prevalence was surprisingly low in both groups. Adjustments were also made for anxiety and depression, but also these conditions tend to be under-reported using only DSM-9 CM codes.

Reply: Lacking of some important confounding factors, such as BMI and smoking history, is the major limitation of our study. The under-reported prevalence of some related disease is also a limitation. We have listed them in our limitation section.

Comment: The authors refer to Sivertsen's paper from the Norwegian HUNT study published in 2014. However, I think that another, more recent and highly relevant publication from the HUNT study by Brumpton et al: Prospective study of insomnia and incident asthma in adults: the HUNT study (ERJ, 2017) also should be referred to.

Reply: Thank you for providing the new reference for our study. We have reviewed the article and added it to our reference.

Reviewer: 2

Reviewer Name: Dr Himender Makker

Institution and Country: University College London Hospital, London UK

Comment: In this prospective cohort study the authors have investigated insomnia as a risk factors for new onset asthma in Taiwanese population and found that patients with insomnia had almost twice the risk of developing asthma.

Approximately 50,000 patients with newly diagnosed insomnia and twice the number of matched control were followed up by four years for new onset asthma.

The diagnostic tool used for diagnosis of insomnia is not stated except that it was diagnosed during one hospital admission and three outpatient visits within a year. Insomnia is a syndrome ranging from acute transient sleep disturbance to chronic sleep disruption.

Similarly the basis for new onset asthma during follow up is not explained except that it was diagnosed during one hospital admission or three clinic visits. Prevalence of atopy related disorders (allergic rhinitis, atopic dermatitis and urticaria) was significantly higher in insomnia population than control group. This could influence diagnosis of new onset asthma in insomnia population. Authors need to discuss possible effect of above differences in diagnosing new onset asthma-despite

Reply: Thanks for the reviewer's comment. The definition of insomnia diagnosis was used the ICD-9-CM codes according to the previous published papers. Therefore, we think the definition of insomnia diagnosis may be acceptable.

Chen, P. J., Huang, C. L. C., Weng, S. F., Wu, M. P., Ho, C. H., Wang, J. J., ... & Hsu, Y. W. (2017). Relapse insomnia increases greater risk of anxiety and depression: evidence from a population-based 4-year cohort study. *Sleep Medicine*.

Huang, C. L. C., Weng, S. F., Wang, J. J., Hsu, Y. W., & Wu, M. P. (2015). Risks of treated insomnia, anxiety, and depression in health care-seeking physicians: a nationwide population-based study. *Medicine*, 94(35).

Chuang, Y. C., Weng, S. F., Hsu, Y. W., Huang, C. L. C., & Wu, M. P. (2015). Increased risks of healthcare-seeking behaviors of anxiety, depression and insomnia among patients with bladder pain

syndrome/interstitial cystitis: a nationwide population-based study. *International urology and nephrology*, 47(2), 275-281.

Wu, M. P., Lin, H. J., Weng, S. F., Ho, C. H., Wang, J. J., & Hsu, Y. W. (2014). Insomnia Subtypes and the Subsequent Risks of Stroke. *Stroke*, 45(5), 1349-1354.

Moreover, we understand the prevalence of atopic diseases was higher in the insomnia group. However, the incidence asthma was still significant higher in the insomnia group after full adjustment.

Comment: Authors need to provide more details for inclusion and exclusion criteria.

Reply: Thanks for reviewer's comment. We add a flowchart for more clearly description.

Comment: Authors have not clarified if patients with asthma were actively excluded from case and control group and basis of exclusion.

Reply: Thanks for reviewer's comment. We add a flowchart for more clearly description.

Comment: Insomnia is dynamic disease and it might have changed /resolved in insomnia group and might have developed in control group during follow up-were above changes monitored during follow up.

Reply:

Thank you for your comments. We understand that the Insomnia is dynamic disease, so the control group was selected as patients without any insomnia records. According to your advice, we revised the relevant part in the manuscript. Your questions are answered as follows.

- (1) All patients diagnosed with asthma (ICD-9-CM code: 493) before insomnia were excluded.
- (2) We excluded all patients with any diagnosis of insomnia during year 2002 to 2011 in the control group.

Reviewer: 3

Reviewer Name: Malvika Sagar

Institution and Country: Mc Lane Children's Hospital/Baylor Scott and White

It is a very large, well written study where authors have addressed a very important question-is insomnia a risk factor for asthma.

I have few minor questions related to the paper.

1. Under the material and method section lines 41-44, Patients diagnosed with asthma before insomnia were excluded. It is possible that the patients with symptoms of asthma and not diagnosed yet may have been included. You may add that in limitation of the study.

Reply: We have added this to our limitation section.

2. Figure 1: Please consider rephrasing the legend to 'The probability of developing asthma in insomnia and control group'.

Reply: We have revised the legend.

3. Sometimes in adult population COPD can be misdiagnosed as asthma and vice versa. The same patient may have been diagnosed with COPD and asthma. Could you look into that?

Reply: Thanks for the reviewer's comment. We all agree that the COPD and asthma could be misdiagnosed each other. Thus, we did a sensitivity analysis for COPD and asthma, and we found the similar risk between insomnia and compared cohorts.

Insomnia(n = 48,871)	Control(n = 97,742)	Hazard Ratio (95% C.I.)	
Asthma 424(0.87)	409(0.42)	1.89 (1.64-2.17)	
Asthma only 316(0.65)	297(0.30)	1.97(1.68-2.32)	
Asthma with COPD 108(0.22)	112(0.11)	1.89(1.45-2.48)	

4. Do you have data on how many patients with insomnia were treated? It would be interesting to note if there were any differences in outcome in those insomnia patients who were treated versus those who were not. This may add weight to your conclusion that proper treatment of insomnia patients might help prevent the progress of airway inflammation.

Reply: Thanks for your suggestion. The effect of medication is an interesting issue, but this information was not available based on our current database. We add this issue into limitation section and will conduct further study to clarify according to your suggestion.

5. Your sample size is very large. However, for the sake of completeness, did you do the power calculation?

Reply: Thanks for reviewer's comment. The description of statistical power has added in the section of statistical analysis.

" The detectable hazard ratio of 1.02 between Insomnia group and compared controls was estimated at 90% statistical power and the probability of type I error at 0.05."

VERSION 2 – REVIEW

REVIEWER	Fredrik Sundbom Uppsala University, Sweden
REVIEW RETURNED	18-Oct-2017

GENERAL COMMENTS	The limitations of the study have now been more clearly addressed.
--

REVIEWER	Dr HK Makker UCLH London, UK
REVIEW RETURNED	20-Oct-2017

GENERAL COMMENTS	Authors have made changes suggested and outline weakness and strength of the study
--

REVIEWER	Malvika Sagar Mc Lane Children's Hospital/Baylor Scott and White
REVIEW RETURNED	29-Oct-2017

GENERAL COMMENTS	All my questions and comments have been appropriately addressed in the letter and incorporated in the manuscript. Thank You.
--

VERSION 2 – AUTHOR RESPONSE

Reviewer: 1

Reviewer Name: Fredrik Sundbom

Institution and Country: Uppsala University, Sweden

Please state any competing interests: none declared

Please leave your comments for the authors below

Comment: The limitations of the study have now been more clearly addressed.

Reply: Thanks for your comment.

Reviewer: 2

Reviewer Name: Dr HK Makker

Institution and Country: UCLH, London, UK

Please state any competing interests: NONE

Please leave your comments for the authors below

Comment: Authors have made changes suggested and outline weakness and strength of the study

Reply: Thanks for your comment.

Reviewer: 3

Reviewer Name: Malvika Sagar

Institution and Country: Mc Lane Children's Hospital/Baylor Scott and White

Please state any competing interests: None

Please leave your comments for the authors below

Comment: All my questions and comments have been appropriately addressed in the letter and incorporated in the manuscript. Thank You.

Reply: Thanks for your comment.